# Nrf1 is not a direct target gene of SREBP1, albeit both are integrated into the rapamycin-responsive regulatory network in human hepatoma cells

Keli Liu[1,2,3], Shaofan Hu[1,2,3], Lu Qiu[1,3,4], Meng Wang[1,3], Zhengwen Zhang[5], Guiyin Sun[2], Yiguo Zhang[2,3] *

1 Bioengineering College, Chongqing University, Shapingba District, Chongqing, China, 2 Chongqing University Jiangjin Hospital, School of Medicine, Chongqing University, Jiangjin District, Chongqing, China, 3 The Laboratory of Cell Biochemistry and Topogenetic Regulation, College of Bioengineering, Chongqing University, Shapingba District, Chongqing, China, 4 School of Life Sciences, Zhengzhou University, Zhengzhou, Henan, China, 5 Laboratory of Neuroscience, Institute of Cognitive Neuroscience and School of Pharmacy, University College London, London, England, United Kingdom

* yiguozhang@cqu.edu.cn

**Data Availability Statement:** All relevant data are within the paper and its Supporting information files.

## Abstract

The essential role of protein degradation by ubiquitin-proteasome system is exerted primarily for maintaining cellular protein homeostasis. The transcriptional activation of proteasomal genes by mTORC1 signaling depends on Nrf1, but whether this process is directly *via* SREBP1 remains elusive. In this study, our experiment evidence revealed that Nrf1 is not a direct target of SREBP1, although both are involved in the rapamycin-responsive regulatory networks. Closely scrutinizing two distinct transcriptomic datasets unraveled no significant changes in transcriptional expression of Nrf1 and almost all proteasomal subunits in either *siSREBP2*-silencing cells or *SREBP1*$^{-/-}$MEFs, when compared to equivalent controls. However, distinct upstream signaling to Nrf1 dislocation by p97 and its processing by DDI1/2, along with downstream proteasomal expression, may be monitored by mTOR signaling, to various certain extents, depending on distinct experimental settings in different types of cells. Our further evidence has been obtained from *DDI1*$^{-/-}$(*DDI2*$^{insC}$) cells, demonstrating that putative effects of mTOR on the rapamycin-responsive signaling to Nrf1 and proteasomes may also be executed partially through a DDI1/2-independent mechanism, albeit the detailed regulatory events remain to be determined.

## 1. Introduction

The normal homeostasis must have to be maintained in all healthy life forms, due to homeostasis robustness, plasticity and resilience ensuring that their structural organization, physiological function and biological behavior are being properly performed and perpetuated at a stable, robust steady-state [1]. Conversely, defects in the maintenance of cell fitness and homeostasis have emerged as an underlying feature of a vast variety of pathologies, such as

**Funding:** This study was funded by the National Natural Science Foundation of China (NSFC, with two project grants 81872336 and 82073079) awarded to Prof. Yiguo Zhang. The funders had no role in study design, data collection and analysis, decision to publish, or preparation of the manuscript.

**Competing interests:** The authors have declared that no competing interests exist.

cancer, senescence and aging-related diseases [2–4]. Of note, protein homeostasis (i.e. proteostasis) is preserved or not, depending on a steady balance between protein synthesis and turnover [5, 6]. Interestingly, as reported by Manning's group [7], such a finely-programmed balance between protein synthesis and degradation was coordinately regulated by the mechanistic target of rapamycin complex 1 (mTORC1), a central kinase that is generally activated by cell growth- and proliferation signaling to trigger protein translation [8]. Besides, they had also shown that mTORC1 signaling activates transcriptional expression of nuclear factor erythroid 2-related factor 1 (Nrf1 with multiple isoforms, encoded by *NFE2L1*) directly by sterol regulatory element-binding protein 1 (SREBP1) [7, 9], although SREBP1 has been commonly accepted as a key control of lipid synthesis and membrane homeostasis, even in response to the rather multifaceted mTOR signaling [10–12].

Intriguingly, in the present study, our evidence has been presented revealing that Nrf1 is not a direct target of SREBP1, albeit they are involved in the rapamycin-responsive regulatory networks. Further experiments are designed to determine whether p97, DDI1 (also called VSM1 (v-SNARE binding protein-1)), which is a highly conserved aspartyl protease among all eukaryotes from yeast to human [13–15] and DDI2 (only present in vertebrates [16]) exert the putative rapamycin-responsive effects on Nrf1 processing and activity to regulate the proteasomal expression. Moreover, the human HepG2-derived $DDI1^{-/-}(DDI2^{insC})$ cell line and tumor xenograft model have also been established herein, to further elucidate the rapamycin-responsive effects on Nrf1 and cognate proteasomes.

## 2. Materials and methods

### 2.1 Cell culture and treatments

HepG2 and HL7702 cell lines were grown in DMEM supplemented with 5 mM glutamine, 10% (v/v) fetal bovine serum (FBS), 100 units/ml of either penicillin or streptomycin, in a 37°C incubator with 5% $CO_2$. Additional HepG2-derived cell lines with the knockout of $DDI1^{-/-}$ were herein established by CRISPR-editing of *DDI1* with specific gRNA (S1 Table in S2 File). The authenticity of $DDI1^{-/-}$ cells had been confirmed by its authentication analysis. Thereafter, experimental cells were transfected with a Lipofectamine 3000 mixture with indicated plasmids or siSREBP1 (with a pair of sequences, S1 Table in S2 File) for 8 h, and allowed for recovery from transfection in a fresh medium for 24 h before being experimented. Additional cells were treated with the mTOR inhibitor rapamycin (RAPA, 20 to 200 nM) or proteasomal inhibitor MG132 (1 to 10 μM) for different time periods (4, 16 or 24h).

### 2.2 Expression constructs

An expression construct for human SREBP1 was made by cloning its full-length cDNA sequence into the pcDNA3 vector, with a pair of its forward and reverse primers (S1 Table in S2 File), which were synthesized by Sangon Biotech Co. (Shanghai, China). Another expression plasmid of Nrf1 was reported previously [17]. The fidelity of all these constructs was confirmed to be true by sequencing.

### 2.3 Luciferase reporter assay

After experimental cells ($1.0 \times 10^5$) were allowed for growth in each well of the 12-well plates to reach 80% confluence, they were co-transfected with a Lipofectamine 3000 mixture with *pNrf1-luc* or *pNrf2-luc* established by Qiu *et al.* [18], plus other expression plasmids. In this dual reporter assay, the *Renilla* expression by pRL-TK served as an internal control for transfection efficiency. The resulting data were normalized from at least three independent

experiments, each of which was performed in triplicate, and thus shown as a fold change (mean ± S.D) relative to the control values.

## 2.4 Quantitative real-time PCR

About 500 ng of total RNAs from experimental cells were subjected to reverse-transcriptase reaction to generate the first strand of cDNA. The newly-synthesized cDNA was used as the template for quantitative PCR in the Master Mix, before being deactivated at 95˚C for 10 min, and amplified by 40 reaction cycles of annealing at 95˚C for 15 s and then extending at 60˚C for 30 s. The final melting curve was validated to examine the amplification quality. While β-actin mRNA level was employed as an optimal internal standard control, target gene expression levels were determined by quantitative real-time PCR, as described previously [19], with each pair of the indicated primers (S1 Table in S2 File). The resulting data were shown a fold change (mean ± S.D) relative to the control values.

## 2.5 Western blotting with distinct antibodies

Total cell lysates in a lysis buffer (0.5% SDS, 0.04 mol/L DTT, pH 7.5) with protease and phosphatase inhibitors (each of cOmplete and PhosSTOP EASYpack tablets in 10 ml buffer), were denatured immediately at 100˚C for 10 min, sonicated sufficiently, and diluted in 3× loading buffer (187.5 mmol/L Tris-HCl, pH 6.8, 6% SDS, 30% Glycerol, 150 mmol/L DTT, 0.3% Bromphenol Blue) at 100˚C for 5 min. Subsequently, equal amounts of protein extracts were subjected to separation by SDS-PAGE containing 4–15% polyacrylamide, and then visualization by Western blotting with distinct antibodies as indicated (S1 Table in S2 File). Some of the blotted membranes were stripped for 30 min and re-probed with additional primary antibodies. Therein, β-actin or GAPDH served as an internal control to verify equal loading of proteins.

## 2.6 Subcutaneous tumor xenograft model

Mouse xenograft models were made by subcutaneously heterotransplanting human HepG2 or derived $DDI1^{-/-}$ cells. Briefly, equal amounts of cells ($1 \times 10^7$) growing in the exponential phase was suspended in 0.1 ml of serum-free medium and then inoculated subcutaneously at a single site in the right upper back region of male nude mice (BALB/C nu/nu, 4–6 weeks, 18 g). The procedure of injection into all the mice was completed within 30 min. Thereafter, the formation of murine subcutaneous tumor xenografts was successively observed until they were sacrificed. These transplanted tumors were excised immediately after being executed, and also calculated in size by a standard formulate ($V = ab^2/2$). All mice were maintained under standard animal housing conditions with a 12-h dark cycle and also allowed access *ad libitum* to sterilized water and diet, according to the institutional guidelines for care and use of laboratory animals with a license SCXK (JING) 2019–0010. All experimental procedures were approved by the Ethics Committee of Chongqing Medical University.

## 2.7 Pathohistology with H&E staining

The xenograft tumor tissues were immersed in 4% paraformaldehyde overnight and then transferred to 70% ethanol. In processing cassettes, tumor tissues were dehydrated by a serial alcohol gradient and then embedded in paraffin wax blocks, before being sectioned into a series of 5-μm-thick slides. Subsequently, the tissue sections were de-waxed in xylene, rehydrated through decreasing concentrations of ethanol and washed in PBS, before being stained

by routine hematoxylin and eosin (H&E) and visualized by microscopy. The resulting images were photographed herein.

## 2.8 Statistical analysis

Significant differences were statistically determined using the Student's t-test and Multiple Analysis of Variations (MANOVA), except for somewhere indicated. The data are here shown as a fold change (mean ± S.D.), each of which represents at least three independent experiments that were each performed in triplicate.

# 3. Results and discussion

## 3.1 Transcriptional expression of Nrf1 is unaffected by SREBP1

Based on the data obtained from chromatin immunoprecipitation (ChIP) [7] and microarray [20, 21], Manning's group considered that aberrant activation of mTOR in $TSC2^{-/-}$MEFs led to upregulation of Nrf1 by SREBP1. However, we showed that transcriptional expression of Nrf1 was unaffected by SREBP1 knockdown or overexpression in both HepG2 and 7702 cell lines (Fig 1A to 1D). Almost no changes in transactivation of *Nrf1* or *Nrf2* promoter-driven luciferase reporters (*pNrf1-luc* and *pNrf2-luc*, established by Qiu *et al.* [18]) were determined in the cellular response to rapamycin (RAPA), but they were induced by *tert*-butylhydroquinone (tBHQ, a pro-oxidative stressor) rather than the antioxidant N-Acetyl-L-cysteine (NAC) (Fig 1E to 1H). Such discrepancy may result from a strange disparity in the cytosolic to the nuclear distribution of Nrf1 examined by Manning's lab (S1A Fig in S2 File, cropped from their extended data [7]). This is due to the authors showing abnormal accumulation of a peculiar nuclear Nrf1 isoform with a higher molecular weight than its cytosolic isoform, which is never recovered by all relevant subcellular fractionation experiments (cropped in S1B to S1D Fig in S2 File, as reported by our and other groups) [22–25]. Such strange nuclear Nrf1 accumulation revealed by Manning's group seems to challenge against the well-established spatiotemporal character of this ER-localized transcription factor [26, 27].

Intriguingly, further examination showed that abundances of all Nrf1 isoforms were marginally enhanced by silencing of SREBP1 (Fig 2A, *cf. a2 vs a1*). Accordingly increased mRNA levels of Nrf1-target *PSMB6*, *PSMB7* and *PSMB5* (encode the core proteasomal ß1, ß2 and ß5 subunits, respectively [28]) were accompanied by enhanced protein abundances of PSMB6, PSMB7 and slightly PSMB5, concomitantly with SREBP1 knockdown (Fig 2B and 2D). The difference between protein and mRNA levels of PSMB5 (as a key housekeeper in the ubiquitin-proteasomal system) suggests that its protein stability may be finely tuned by the proteasomal feedback regulatory loop, facilitating proteostasis maintenance. Conversely, phosphorylated S6 kinase 1 (pS6K1, required for protein synthesis) protein and mRNA expression levels were significantly down-regulated by *siSREBP1* (Fig 2B and 2D). Thereby, SREBP1 may be directionally responsible for regulating protein synthesis and degradation.

Next, an examination of *siSREBP1*'s effects on the upstream regulators of Nrf1 revealed that evident abundances of DDI1 and DDI2 were increased (Fig 2C). This was accompanied by modest decreases in p97/VCP protein and mRNA levels, while the ER-resident E3 ligase Hrd1 was almost unaffected by silencing of SREBP1 (Fig 2C and 2D). Further scrutinizing two distinct transcriptomic datasets (at https://www.ncbi.nlm.nih.gov/geo/query/acc.cgi?acc= GSE93980 and = GSE90571) unraveled that, though putative SREBP1-binding sites exist in both the promoter region of Nrf1 and its first exon (S2 Fig in S2 File), no significant changes in transcriptional expression of Nrf1 and other homologous factors (Fig 2E and 2F), as well as almost all proteasomal subunits (S3A and S3B Fig in S2 File), were determined in either

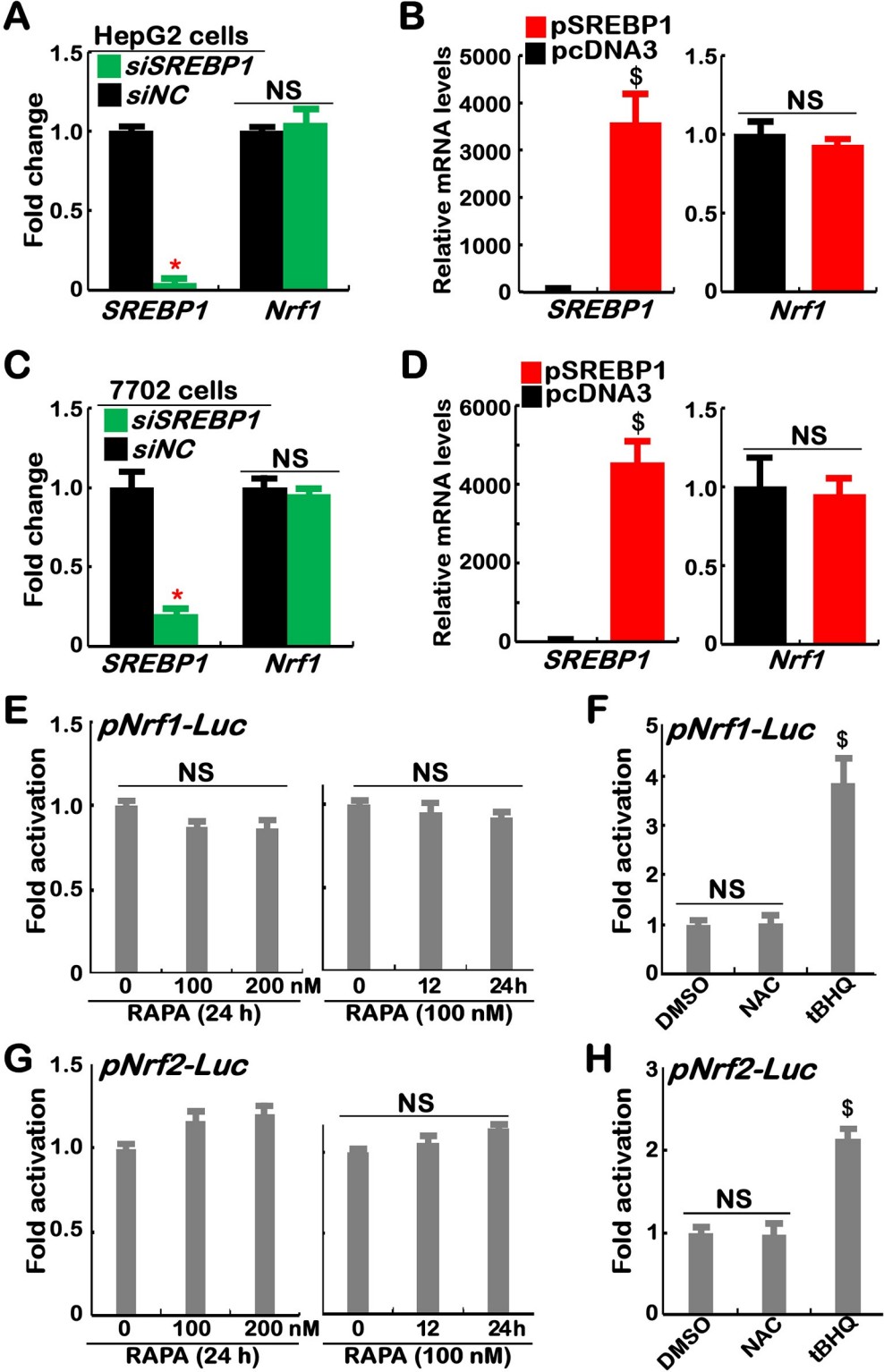

**Fig 1. Transcriptional expression of Nrf1 and its reporter is unaffected by SREBP1 or rapamycin.** (A to D) Two cell lines of HepG2 (*A*, *B*) and L7702 (*C*, *D*), that had been transfected with: (*A*, *C*) siNC (a negative control) or siSREBP1; (*B*, *D*) a pSREBP1 expression construct or empty plasmid, were subjected to real-time qPCR analysis of mRNA expression levels of *SREBP1* and *Nrf1* (n = 3×3; with significant decreases (*, $p < 0.01$), significant increases ($, $p < 0.01$), or no significances (NS)). (E to H) HepG2 cells, that had been transfected with *pNrf1-luc* (*E*, *F*) or *pNrf2-luc*

(*G, H*) reporters, along with *pRL-TK* (an internal control) and then treated for 24 h with rapamycin (RAPA, at 0, 100 or 200 nM) (*E, G*), NAC (10 mM) or tBHQ (50 μM) (*F, H*), were subjected to an assay of dual-luciferase activity (n = 3×3) with significant increases ($, $p<0.01$) or no significances (NS). All the results representing at least three independent experiments, each of which was performed in triplicates, were determined as fold changes (mean ± S.D.) relative to equivalent controls.

*siSREBP2*-silencing PANC-1 cells or *SREBP1*$^{-/-}$MEFs, when compared to their wild-type controls. Collectively, these lines of evidence together demonstrate that Nrf1 is not a direct target of SREBP1, albeit its indirect effects on upstream signaling to Nrf1 cannot be ruled out.

## 3.2 Discrete effects of rapamycin on the signaling to Nrf1 and proteasome

It was, to my surprise, found that stimulation of HepG2 cells by feeding 10% FBS after 10-h free-serum starvation caused significant decreases in mRNA expression levels of Nrf1 and SREBP1, but their protein abundances were strikingly increased to varying extents (Fig 3A *vs* 3B and 3C), and markedly diminished or abolished by rapamycin (20 nM, Fig 3B and 3C). Similar results were obtained for S6K (Fig 3C and 3D). Of note, mRNA expression of S6K, but not Nrf1 or SREBP1, was reversed and increased by rapamycin (Fig 3D). Together, these demonstrate that mTOR is likely involved in at least two different mechanisms for regulating Nrf1 and SREBP1 at mRNA and protein expression levels, which are distinctive from controlling its downstream S6K1 by potential 'bounce-back' response to mTOR inhibitor.

Further experimental evidence showed that Nrf1-target PSMB5, PSMB6 and PSMB7 (Fig 3B), as well as the upstream signaling DDI1, DDI2, p97 and Hrd1 (Fig 3E) were significantly upregulated by feeding FBS, of which all those except Hrd1 were also inhibited by rapamycin. However, their mRNA expression levels were down-regulated or unaffected by FBS, but also partially reversed or event enhanced by rapamycin (Fig 3F, *cf. left vs right panels*). Of note, mRNA expression of *PSMB6* and *DDI2* was unaltered or down-regulated by FBS, respectively, but both were also significantly augmented by rapamycin. Overall, these results further indicate that key upstream and downstream signaling molecules of Nrf1 were, to some certain extent, influenced by mTOR involved in distinct hierarchical mechanisms.

## 3.3 Alteration in the putative processing of Nrf1 in *DDI1/2*-deficient cells

Since Nrf1 and *C. elegans* SKN-1A are activated by DDI1 in the proteasomal 'bounce-back' response [14, 17, 19], we established a *DDI1*$^{-/-}$cell line by CRISPR-editing with specific gRNA (S4A Fig in S2 File). Further examination of *DDI1*$^{-/-}$cells by its DNA sequencing, real-time qPCR and Western blotting revealed that two overlapping nucleotide segments of *DDI1* were deleted from its two alleles (Fig 4A and S4B Fig in S2 File), but the remnant mRNA levels were expressed (Fig 4B and 4C). This implies there may exist alternative mRNA-splicing and in-frame translation start sites to yield two isoforms with distinct molecular weights, as described in yeast DDI1 [29]. In addition to *DDI1*$^{-/-}$, an extra-cytosine base was inserted in the open reading of *DDI2* (S4C Fig in S2 File), thus recalled *DDI1*$^{-/-}$(*DDI2*$^{insC}$) collectively. This should be a result of *DDI1*-recognized gRNA targeting the highly conserved sequence of *DDI2* (S4A Fig in S2 File).

Assessment of subcutaneous tumor xenograft mice unraveled that no differences in *in vivo* tumorigenesis and tumor growth of *DDI1*$^{-/-}$(*DDI2*$^{insC}$) cells were observed when compared to those of its parent wild-type cells (Fig 4D and 4E). Also, no obvious changes in their tumor pathohistological sections were shown (in Fig 4F). Such DDI1/2-deficient cells were then treated with distinct concentrations of MG132 and subjected to determination of

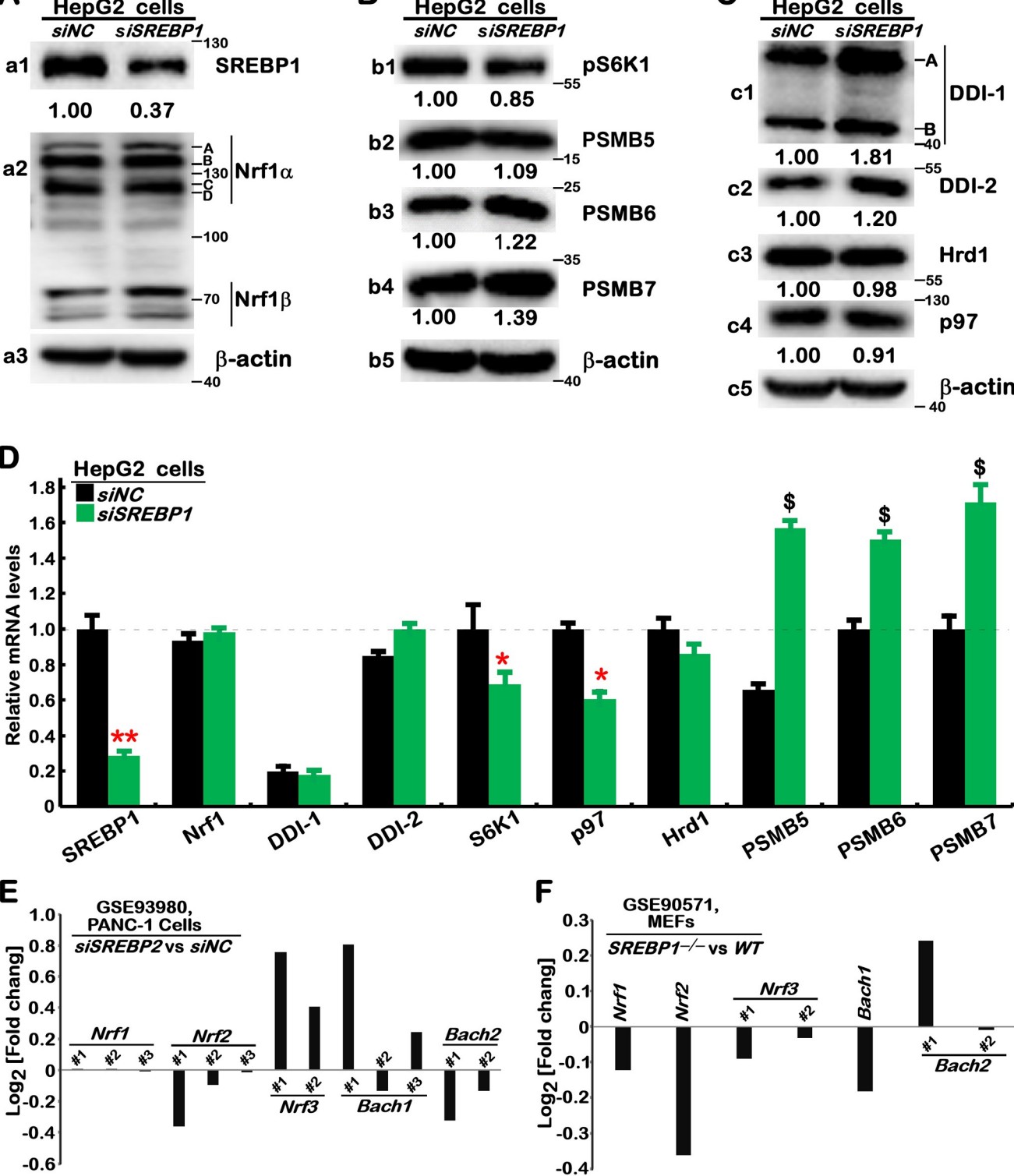

**Fig 2. The upstream signaling to Nrf1 and proteasome are to no or fewer degrees, affected in *SREBP1*-deficient cells.** (A to C) HepG2 cells were transfected with siNC or siSREBP1 for 24 h and then subjected to Western blotting with those indicated antibodies. The intensity of immunoblots representing each protein was quantified by the Quantity-One software and shown on the bottom. (D) The mRNA levels of those examined genes were determined by real-time qPCR and shown as fold changes (mean ± S.D. n = 3×3) with significant decreases (*, $p < 0.01$) or significant increases ($, $p < 0.01$) relative to equivalent controls. These results are representative of at least three independent experiments, each of which was performed in triplicates. (E, F)

No significant changes in transcriptional expression of Nrf1 and other homologous factors were determined by transcriptomic sequencing of *siSREBP2 vs siNC* (https://www.ncbi.nlm.nih.gov/geo/query/acc.cgi?acc=GSE93980) in PANC-1 cells (*E*), as well as *SREBP1⁻/⁻vs Wild-type* MEFs (https://www.ncbi.nlm.nih.gov/geo/query/acc.cgi?acc=GSE90571) (F).

putative effects of $DDI1^{-/-}(DDI2^{insC})$ on the processing of Nrf1. As anticipated, the results revealed that processed Nrf1 isoforms-C/D were significantly reduced, but its full-length glycoprotein-A and deglycoprotein-B were almost unchanged following treatment of $DDI1^{-/-}(DDI2^{insC})$ with a lower dose (1 μM) of MG132 when compared to those measured from wild-type cells (Fig 4G, *middle two lanes*). By sharp contrast, a higher dose (10 μM) of MG132 treatment of $DDI1^{-/-}(DDI2^{insC})$ caused Nrf1 isoforms-C/D to be further diminished or abolished, but its full-length proteins-A/B were not augmented, when compared to their wild-type controls (Fig 4G, *right two lanes*). These demonstrate a requirement of 26S proteasome for DDI1/2-directed proteolytic processing of Nrf1 because the stability of both proteases *per se* is also controlled by ubiquitin-proteasome pathways [30, 31]. However, endogenous Nrf1 isoforms-A/B was marginally reduced in untreated $DDI1^{-/-}(DDI2^{insC})$ cells, where Nrf1 isoforms-C/D were rather faint to be distinguishable from wild-type controls (Fig 4G, *left two lanes*). This implies that Nrf1α-derived isoforms may be much unstable to be rapidly destructed, but shorter isoforms Nrf1$^{ΔN}$, Nrf1ß and Nrf1γ were unaffected, in *DDI1/2*-deficient cells (Fig 4G, *left two lanes*). Such seemingly-contradictory data, showing no increased full-length Nrf1 isoforms-A/B in $DDI1^{-/-}(DDI2^{insC})$ cells, suggest that ER membrane-associated protein degradation and/or autophagy [32] may also be triggered in possibly 'bounce-back' response to *DDI1/2* deficiency.

Intriguingly, wild-type DDI1 and its short isoform in $DDI1^{-/-}(DDI2^{insC})$ cells were not enhanced, but slightly reduced by treatment of 1 μM or 10 μM MG132 for 4 h (Fig 4H, *h1*), and the reduced abundances were further decreased as treatment time was extended to 24 h (Fig 4H, *h4*) when compared with their untreated controls. By contrast, wild-type DDI2 and its remaining protein in $DDI1^{-/-}(DDI2^{insC})$ cells were largely unaffected by 24-h treatment of 1 μM or 10 μM MG132 (Fig 4H, *h5*), but after 4-h treatment of cells, they became marginally reduced by 1 μM MG132, and also rather augmented by 10 μM MG132 (Fig 4H, *h2*). Such distinct effects of this proteasomal inhibitor on DDI1 and DDI2 demonstrate that both protease stability may be governed through different mechanisms, albeit these details remain to be elucidated.

### 3.4 *DDI1/2*-deficient effects on the rapamycin-responsive signaling to Nrf1

Herein, we also found marked decreases in the ectopic expression of Nrf1α-derived isoforms in $DDI1^{-/-}(DDI2^{insC})$ cells, when compared to wild-type cells (Fig 5A, *a1* and *a3*). Further comparisons revealed that Nrf1-target proteasomal subunits PSMB5, PSMB6, PSMB7 were accordingly downregulated to considerably lower degrees in *DDI1/2*-deficient cells (Fig 5A, *a5 to a7*). Also, *DDI1/2* deficiency enabled for significant downregulation of p97 (acting as key upstream signaling to Nrf1) (Fig 5B, *b2*), in addition to DDI2 and Nrf2 (Fig 5B, *b1* and *b4*). Rather, a relative increase in basal expression of Keap1 (as a negative regulator of Nrf2 [33]) was determined in $DDI1^{-/-}(DDI2^{insC})$ cells (Fig 5B, *b5*). Further real-time qPCR data unraveled various decreases in mRNA expression levels of *p97*, *DDI2*, *PSMB5*, *and PSMB7*, but not of *PSMB6*, *Nrf1* or *Nrf2* in *DDI1/2*-deficient cells (Fig 5C).

As such, abundances of endogenous Nrf1α-derived proteins were significantly increased by feeding 10% FBS for 12–24 h, after 10-h free-serum starvation of $DDI1^{-/-}(DDI2^{insC})$ cells, and these FBS-stimulated increases were almost completely abolished by 20 nM rapamycin

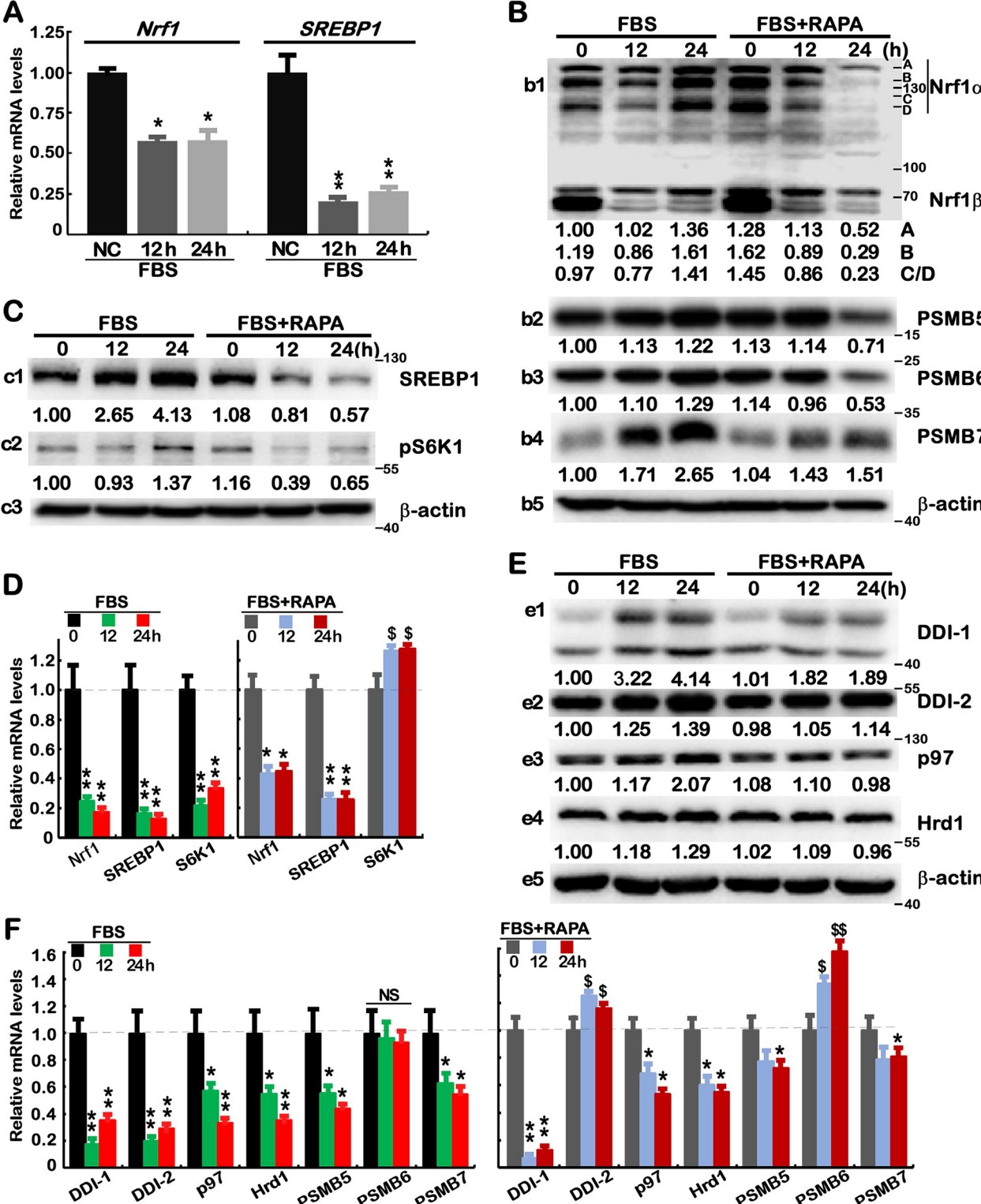

**Fig 3. Distinct effects of rapamycin on FBS-altered expression of SREBP1, Nrf1 and relevant signaling molecules.** (A) HepG2 cells that had been starved in a serum-free medium for 10 h and then stimulated for 12 or 24 h by feeding 10% FBS, were subjected to real-time qPCR analysis of Nrf1 and SREBP1 at mRNA expression levels. The results were shown as fold changes (mean ± S.D. n = 3×3) with significant decreases (*, $p < 0.01$; **, $p < 0.001$) relative to the negative controls (NC, with no FBS treatment). (B to F) The free-serum starved HepG2 cells were treated with 10% FBS alone or plus 20 nM RAPA for 0, 12, 24 h, before being subjected to Western blotting with indicated antibodies (*B*, *C*, *E*), in which the intensity of

immunoblots was calculated and shown on the bottom, or real-time qPCR analysis of indicated genes at mRNA levels (*D, F*). The results were shown as fold changes (mean ± S.D. n = 3×3) with significant decreases (*, $p<0.01$; **, $p<0.001$), significant increases ($, $p<0.01$; $$, $p<0.001$), or no significant differences (NS), relative to their equivalent controls. These results are representative of at least three independent experiments, each of which was performed in triplicates.

(Fig 5D, *d1*). By contrast, FBS-stimulated protein expression of Nrf2 was partially suppressed by rapamycin (Fig 5D, *d2*). However, mRNA expression levels of *Nrf1* and *Nrf2* were not induced, but rather repressed by FBS, and such repression was partially or completely reversed by rapamycin, as a result of enhanced mRNA expression of *Nrf2* by this mTOR inhibitor (Fig 5F and 5G). These data indicate there exists a feedback negative regulatory circuit between mRNA and protein expression of Nrf1 and Nrf2, during stimulation or inhibition of mTOR.

Further examinations of *DDI1*$^{-/-}$(*DDI2*$^{insC}$) cells uncovered that, upon the absence of DDI1, the remnant DDI2 were still partially enhanced by FBS, and also partially inhibited by rapamycin (Fig 5E). Thereof, FBS-repressed mRNA expression of *DDI2* was fully reversed to a slight increase by rapamycin (Fig 5F and 5G). Similarly, modest stimulation of PSMB5, PSMB6 and PSMB7, abundances by FBS was also partially inhibited by rapamycin (Fig 5E, *e3* to *e5*), but their FBS-repressed mRNA levels were not reversed by rapamycin (Fig 5F and 5G). Such opposite (stimulatory or inhibitory) effects of mTOR on rapamycin-responsive signaling to Nrf1 and downstream proteasome may partially occur in DDI1/2-deficient cells, implying that a DDI1/2-independent mechanism also accounts for this process.

## 3.5 Discussion

The ubiquitin proteasome system is crucial for protein degradation and homeostasis, whilst Nrf1 is a key regulatory factor for governing the transcriptional expression of all proteasome subunits. Of note, the transcriptional activation of proteasomes by mTORC1 may be in an Nrf1-dependent manner, and the mTORC1 signaling to upregulation of Nrf1-targeted proteasomal expression profiles was also considered to occur directly by SREBP1, as reported by Manning's group in 2014's Nature [7]. However, a core point from Manning's work on the control of proteasomal proteolysis by mTOR [7] was first challenged by Goldberg's group, arguing that their methodology to measure the rates of protein degradation (labelled by $^{35}$S-Met/Cys rather than by $^3$H-Phe) and the former labelled-protein pulse-chase experimental data appear questionable [34], because the resulting data were considered to be rather inconsistent with those well-established discoveries [35, 36]. Amongst the best-studied actions of mTORC1 are enhancing protein synthesis, and also inhibiting protein degradation by autophagy and proteasomes [35, 37]. While mTORC1 is inactivated during starvation, an increase in proteolysis and autophagy provides a recycling of amino acids for next protein synthesis and energy production. As such, Manning's group showed that inhibition of mTORC1 activity for 16 h or more resulted in a delayed reduction in overall proteolysis by down-regulating transcriptional expression of proteasomal subunits [7]. Similar results were also obtained from $^3$H-Phe-labelled pulse-chase experiments in their reply to doubt by Goldberg's group [38]. Such discrepant results presented by Manning's and Goldberg's groups are not attributable to nuances in the assays but are inferable due to differences in their chosen culture conditions [38]. A higher dose of rapamycin (300 nM, at least 100-fold more than the $IC_{50}$ for inhibiting mTORC1) was employed [34] to enable two mTOR kinase complexes (i.e., mTORC1 and mTORC2) to be completely blocked in mouse embryonic fibroblasts (MEFs) with genetic loss of tuberous sclerosis complex 2 (*TSC2*$^{-/-}$,

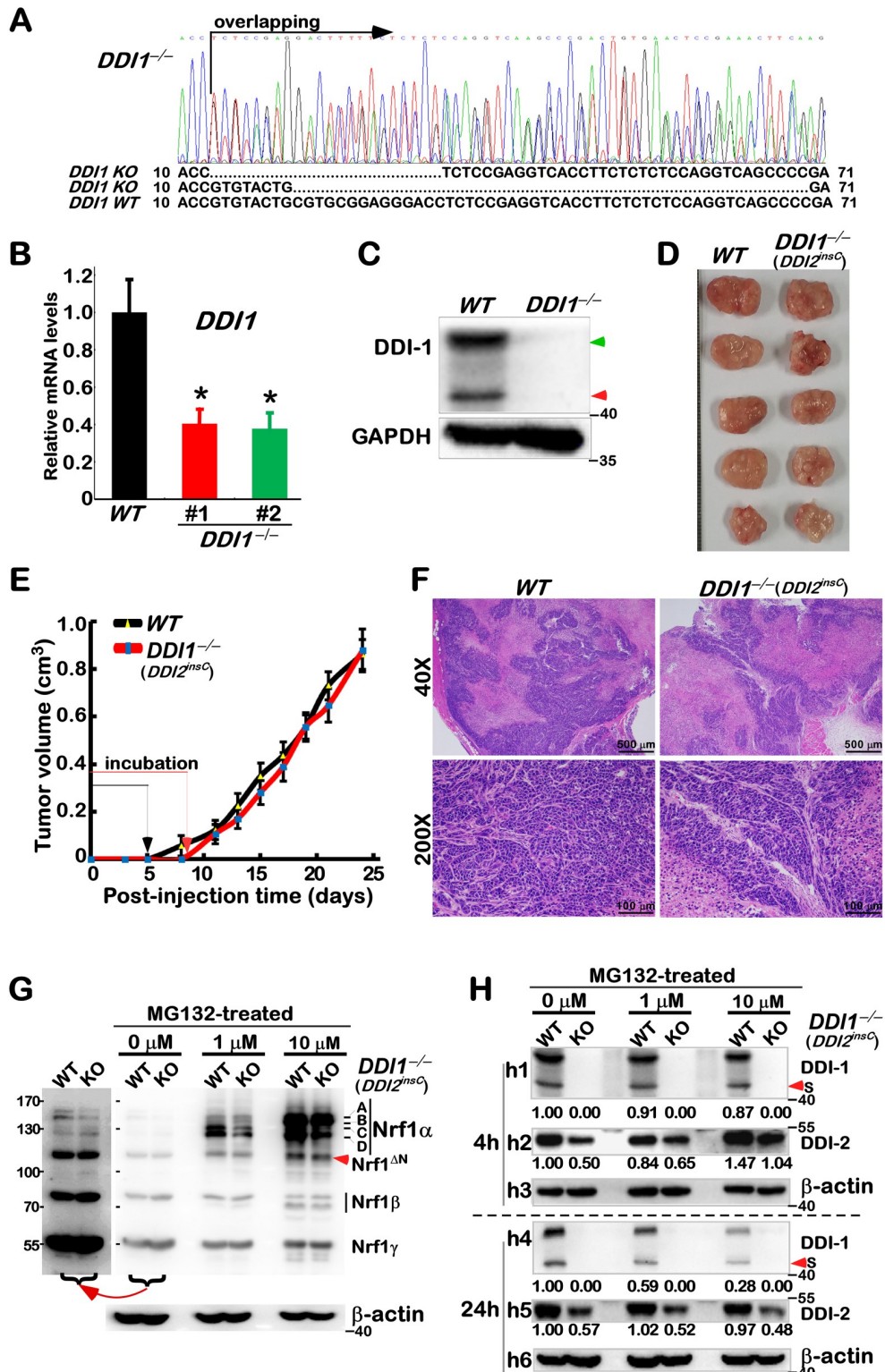

**Fig 4. Changed processing of Nrf1 in *DDI1/2*-deficient cells, but with no different xenograft models.** (A) HepG2-derived *DDI1*$^{-/-}$ cells were initially identified by their genomic DNA-sequencing. The results were shown graphically, along with the alignment of two mutant alleles and wild-type (*WT*). (B, C) In contrast with *WT* cells, *DDI1*$^{-/-}$ cells were further determined by real-time qPCR (*B*, shown by mean ± S.D. n = 3×3; *, *p*<0.01) and Western blotting (*C*), respectively. (D) No different phenotypes of xenograft tumors in nude mice were observed after murine

subcutaneous inoculation of *WT* and *DDI1*$^{-/-}$(*DDI2*$^{insC}$) hepatoma cells. (D) No differences in both tumorigenesis and *in vivo* growth between *WT and DDI1/2*-deficient and xenograft tumors were measured in size every two days, before being sacrificed. The results are shown as mean ± S.D. (n = 5). (F) The pathohistological images were obtained by routine HE staining of the aforementioned xenograft tumor tissues. (G, H) Both lines of *WT* and KO (i.e. *DDI1*$^{-/-}$*DDI2*$^{insC}$) cells were treated with MG132 at 0, 1 or 10 μM for 24 h (*G, H*) or 4 h (*H*), and then subjected to Western blotting with distinct antibodies against Nrf1, DDI1 or DDI2. In addition, a long-term exposed image was cropped from part of the corresponding gel (G). These results are representative of at least three independent experiments, each of which was performed in triplicates.

that is accompanied by aberrant activation of mTOR). By contrast, a much lower dose (20 nM) of rapamycin for treatment of *TSC2*$^{-/-}$ cells grown in the low-serum conditions [7, 38] enabled for specific separation of effects of mTORC1 from mTORC2, as described by [39]. Just under this status, it was found that *TSC2*$^{-/-}$-leading activation of mTORC1, rather than mTORC2, stimulates a transcriptional programme involving SREBP1 and Nrf1, leading to an evident enhancement of proteasome-mediated proteolysis exclusively by Nrf1, but not Nrf2 [7, 9].

Since Manning's work was fantastically well done [7, 38] and has gained nearly 200 citations (from the Web of Science at https://www.webofscience.com), it is worth interrogating why no more further experimental evidence confirming their findings has been provided to date by any other groups, so far as we know. Of particular concern is a key issue arising from Manning's work, which merits reexamination of whether mTORC1 signaling to upregulation of Nrf1-targeted proteasomal expression profiles occurs directly by SREBP1 because this controversial mechanism remains obscure. Here, we found that transcriptional expression of Nrf1 and all proteasomal subunits is almost unaffected by SREBP1 (or SREBP2), but conversely, Nrf1 contributes to negative regulation of SREBP1 involved in lipid metabolism ([40] and this study). Besides, a Yin-Yang relationship between Nrf1 and SREBP2 for maintaining cholesterol homeostasis was elaborately unraveled in Hotamisligil's laboratory [22]. Recently, a mechanistic study by Xu's group [41] has shown that activity of SREBPs is inhibited by promoting degradation of SREBP-cleavage activating protein (SCAP, a central sensor for cholesterol) through the protein-ubiquitin E3 ligase RNF5-dependent proteasomal pathway, in this process whereby this ligase is recruited to the endoplasmic reticulum (ER)-localized transmembrane protein 33 (TMEM33), a direct target of Nrf1. Therein TMEM33 is also a downstream effector of pyruvate kinase isoform 2 (PKM2), which coordinates together with p97/VCP to control the processing of Nrf1 and SREBPs, as well as their bidirectional regulatory ability to dictate lipid metabolism and homeostasis [41].

Because Nrf1 and SREBP1 manifest distinct topobiological behavior around membranes [42], they are endowed with their respective discrepant spatiotemporal partitioning and unique biological functionality to be though exerted, only after being dislocated from the ER into the nucleus so as to regulate distinct sets of target genes. In this study, we have discovered that Nrf1 is not a direct target of SREBP1, albeit both are involved in rapamycin-responsive signaling networks (Fig 6). Of note, the upstream signaling to Nrf1 dislocation by p97 and its processing by DDI1/2, along with downstream proteasomal expression, should be indirectly monitored by potent mTOR signaling networks, to various certain extents, depending on distinct experimental settings in distinct cell types. Therefore, the potential indirect effects of SREBP1 on DDI1/2 and proteasomes cannot be ruled out. Further experimental evidence from *DDI1*$^{-/-}$(*DDI2*$^{insC}$) cells demonstrates that putative effects of mTOR on the rapamycin-responsive signaling to Nrf1 and proteasome may also be executed partially through a DDI1/2-independent mechanism, albeit the detailed regulatory events remain to be elucidated.

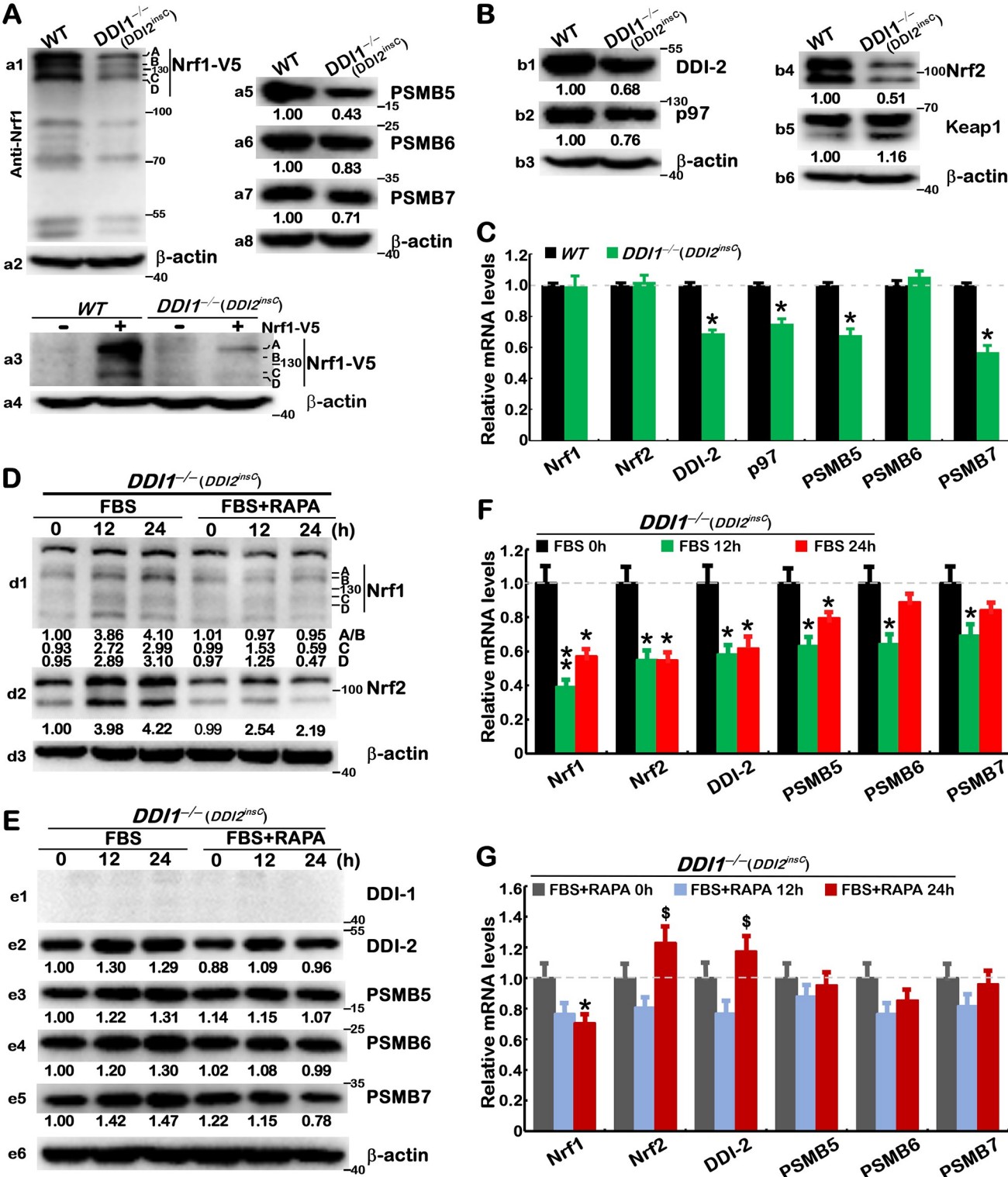

**Fig 5. *DDI1/2*-deficient effects on the rapamycin-responsive signaling to Nrf1 and proteasome.** (A) *WT* and *DDI1⁻/⁻(DDI2ⁱⁿˢᶜ)* cell lines were transfected with an expression construct for Nrf1-V5 (+) or empty pcDNA3 vector (−) and then examined by Western blotting with V5 antibody (*a1, a3*); ß-actin acts as a loading control. Their untransfected cells were also measured by immunoblotting of the core proteasomal subunits PSMB5, PSMB6, and PSMB7 (*a5 to a7*). (B) Further immunoblotting of DDI2, p97, Nrf2 and Keap1 was conducted in untreated *WT* and *DDI1⁻/⁻(DDI2ⁱⁿˢᶜ)* cell lines. (C) Both cell lines were further assessed by real-time qPCR analysis of mRNA expression levels. The results were shown as fold changes (mean ± S.D. n = 3×3) with a significant decrease (*, *p*<0.01) relative to control values. (D to G) The starved *DDI1⁻/⁻(DDI2ⁱⁿˢᶜ)* cells were treated by feeding 10% FBS alone or plus

RAPA (20 nM) for 0, 12 or 24 h, and then subjected to Western blotting with distinct antibodies (*D, E*) and real-time qPCR analysis of mRNA expression (*F, G*). The resulting data were shown as fold changes (mean ± S.D. n = 3×3), with a significant decrease (*, $p<0.01$) or significant increases ($, $p<0.01$) relative to control values. The results are representative of at least three independent experiments, each of which was performed in triplicates.

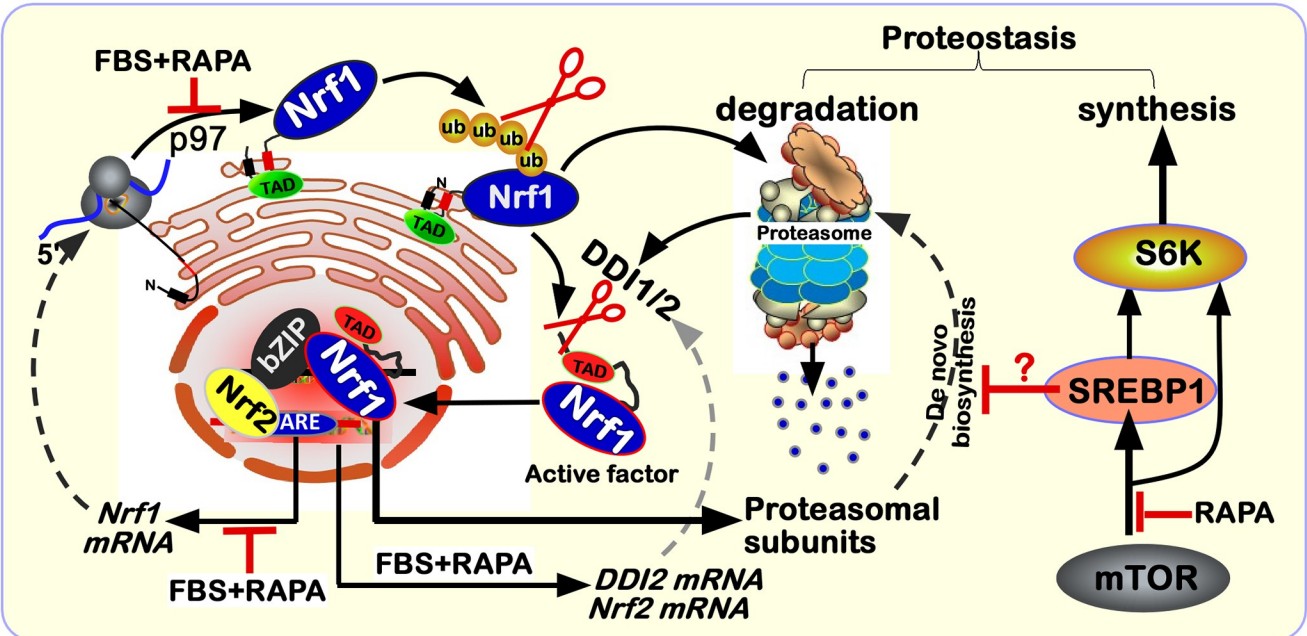

**Fig 6. A model is proposed for a better understanding of the rapamycin-responsive signaling to Nrf1 and proteasomes.** The ER-localized Nrf1 manifests its unique topobiological behavior with specific dislocation by p97 and proteolytic processing by DDI1/2 and proteasomes, to give rise to a mature N-terminally-truncated isoform of this CNC-bZIP factor that mediates proteasomal transcriptional expression. Herein, we found that Nrf1 is not a direct target of SREBP1 (required for lipid and cholesterol metabolism) and Nrf1-target proteasomal transcription is almost not induced, but rather inhibited by SREBP1, although both factors are also integrated into the rapamycin-responsive signaling networks. Besides, differential expression levels of p97, DDI1/2 and Nrf2 may be monitored by mTOR signaling, to various certain extents, depending on the distinct experimental settings in distinct cell types. These detailed regulatory mechanisms should warrant in-depth studies.

## Supporting information

**S1 File. The original files of all western blots results in Figs 2–5.**
(PDF)

**S2 File. A supplement file containing S1-S4 Figs and S1 Table.**
(PDF)

## Acknowledgments

We thank to all those present and past members of Prof. Zhang's laboratory (at Chongqing University, China) for giving critical discussion and invaluable help with this work.

## Author Contributions

**Conceptualization:** Keli Liu, Yiguo Zhang.

**Data curation:** Keli Liu, Shaofan Hu, Lu Qiu.

**Formal analysis:** Keli Liu, Lu Qiu.

**Funding acquisition:** Yiguo Zhang.

**Investigation:** Keli Liu, Shaofan Hu.

**Methodology:** Keli Liu, Shaofan Hu, Lu Qiu, Meng Wang.

**Project administration:** Guiyin Sun, Yiguo Zhang.

**Resources:** Guiyin Sun, Yiguo Zhang.

**Software:** Keli Liu, Meng Wang.

**Supervision:** Keli Liu, Zhengwen Zhang, Yiguo Zhang.

**Validation:** Keli Liu.

**Visualization:** Keli Liu, Meng Wang.

**Writing – original draft:** Keli Liu.

**Writing – review & editing:** Zhengwen Zhang, Yiguo Zhang.

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
