## [Decision Letter · Decision Letter 0]

20 Sep 2023

PONE-D-23-03948Nrf1 is not a direct target gene of SREBP1, albeit both are integrated into the rapamycin-responsive regulatory network in human hepatoma cellsPLOS ONE

Dear Dr. Zhang,

Thank you for submitting your manuscript to PLOS ONE. After careful consideration, we feel that it has merit but does not fully meet PLOS ONE’s publication criteria as it currently stands. Therefore, we invite you to submit a revised version of the manuscript that addresses the points raised during the review process. Both reviewers have identified issues which should be addressed if the authors plan to submit a revised manuscript.  This is particularly true in the case of reviewer 2.

We look forward to receiving your revised manuscript.

Kind regards,

Salvatore V Pizzo

Academic Editor

PLOS ONE

Journal Requirements:

Reviewers' comments:

Reviewer's Responses to Questions

**Comments to the Author**

1. Is the manuscript technically sound, and do the data support the conclusions?

Reviewer #1: Partly

Reviewer #2: Yes

2. Has the statistical analysis been performed appropriately and rigorously? 

Reviewer #1: Yes

Reviewer #2: Yes

3. Have the authors made all data underlying the findings in their manuscript fully available?

Reviewer #1: Yes

Reviewer #2: Yes

4. Is the manuscript presented in an intelligible fashion and written in standard English?

Reviewer #1: Yes

Reviewer #2: Yes

5. Review Comments to the Author

Reviewer #1: 7/23/2023

Editor-in Chief

PLOS ONE

Thank you very much for giving me an opportunity to review Manuscript Number: PONE-D-23-03948 , entitled:” Nrf1 is not a direct target gene of SREBP1, albeit both are integrated into the rapamycin-responsive regulatory network in human hepatoma cells.”

In this manuscript, the authors reported that Nrf1is not a direct target of SREBP1.

Other points are:

Abstract section:

It is suggested that abstract be written more clearly .

Introduction section:

The introduction needs to review more articles.

Also, sentence “Nrf1 is not a direct target of SREBP1, albeit they are involved in the rapamycin-responsive regulatory networks “should be clarified.

Methods section:

It is suggested to write a reference for each method.

Results and Discussion section:

How did the authors conclude that SREBP1 may be directionally responsible for regulating protein synthesis and degradation?

This sentence “Nrf1 is not a direct target of SREBP1, albeit its indirect effects on upstream signaling to Nrf1 cannot be ruled out” needs more explain.

It would be better if revise the conclusion section.

This MS needs revision.

With bests wishes

Dr.Durdi Qujeq

Faculty of Medicine , Babol University of Medical Sciences

Reviewer #2: In this study, the authors discussed about the SREBP1 independent mechanism of Nrf1 response to the mTOR signaling pathway, putative effects of mTOR on the rapamycin-responsive signaling to Nrf1 and proteasomes may also be executed partially through a DDI1/2-independent mechanism. This research provides novel insights into the understanding of Nrf1 and SREBP1. However, before this manuscript can be considered for publication in the journal PLOS ONE, several concerns need to be addressed.

1.Nrf1's processing protein DDI1/2 increase when SREBP1 is knocked down, but Nrf1 does not change, has there been a transform in the nucleocytoplasmic distribution of Nrf1?

2.DDI1 knockout affects the expression of Nrf1 and Nrf2, but there is no change in tumor growth. Is this related to the simultaneous changes in Nrf1 and Nrf2?

3.Is the effect of DDI1 deficiency on the reactive signaling pathway of rapamycin to proteasomes also regulated by the expression of other proteasome subunits?

4.Please check the text for minor errors, e.g., in Figure 5A has an extra letter 'A' above it.

5.(Fig. 4F and Fig. 5)

The resolution of some photographs is too low. Please improve them.

6. PLOS authors have the option to publish the peer review history of their article (what does this mean?). If published, this will include your full peer review and any attached files.

Reviewer #1: No

Reviewer #2: No

---

## [Author Response · Author response to Decision Letter 0]

17 Oct 2023

The point-to-point response to the reviewer’s comment

We are greatly grateful to the reviewers for taking his/her time to read this manuscript and for pointing out some questions. All the points are addressed as shown below. 

Reviewer #1's Comments

Thank you very much for your patient review and instructional suggestions and comments. After careful consideration and summary, we have substantially and seriously revised the text and now we have sorted out all the data and answered as follows:

POINT 1: In this manuscript, the authors reported that Nrf1is not a direct target of SREBP1. Other points are:

Abstract section: It is suggested that abstract be written more clearly.

Response: We agree with this constructive comment. Right now, we have supplemented the sentence “In this study, our experiment evidence showed that knockdown or overexpression of SREBP1 had no significant effect on Nrf1 expression in both normal hepatocytes and hepatoma cells, which revealed that Nrf1 is not a direct target of SREBP1, although both are involved in the rapamycin-responsive regulatory networks ” in the Abstract, so as to make it clearly for a wide readership showed in revised manuscript. 

POINT 2: Introduction section: The introduction needs to review more articles. Also, sentence“Nrf1 is not a direct target of SREBP1, albeit they are involved in the rapamycin-responsive regulatory networks ” should be clarified. 

Response: We are thankful for pointing out this issue. Now, we have reviewed more newly-published articles, such as the 9th、14th and 15th, referenced in this revised paper. Meanwhile, we have supplemented the relevant contents“Nrf1 has also been reported to play a crucial role in maintaining protein homeostasis and regulating cell fate [14, 15], but its mechanism in response to mTOR signaling in the regulation of these biological processes has not been fully elucidated” in the paragraph I of Introduction, in order to illustrate sentence “Nrf1 is not a direct target of SREBP1, albeit they are involved in the rapamycin-responsive regulatory networks ”clearly. In addition, we have rewritten “ the expression level of Nrf1 was not significantly changed in either knockdown or overexpression of SREBP1 in both normal hepatocytes and hepatoma cells, indicating that Nrf1 is not a direct target of SREBP1” for better clarification in the paragraph II.

POINT 3: Methods section: It is suggested to write a reference for each method.

Response: Thanks for your rigorous question a lot. According to your suggestions, as showed in revised version, we have cited the 23th article to support the formula for calculating tumor volume in the part of Subcutaneous tumor xenograft model. Besides, we have also described H&E staining in detail, as the 21th article mentioned in the part of Pathohistology with H&E staining.

POINT 4: Results and Discussion section: How did the authors conclude that SREBP1 may be directionally responsible for regulating protein synthesis and degradation? This sentence “Nrf1 is not a direct target of SREBP1, albeit its indirect effects on upstream signaling to Nrf1 cannot be ruled out” needs more explain. 

Response: We are greatly thankful to you for pointing this out. In our results, when SREBP1 knockdown, both mRNA levels and protein abundances of PSMB6, PSMB7 and PSMB5 were increased, whereas proteasome subunits were necessary for protein degradation. Our results also showed that phosphorylated S6 kinase 1 protein and mRNA expression levels were significantly down-regulated by siSREBP1, but pS6K1 was required for protein synthesis. So we conclude that SREBP1 may be directionally responsible for regulating protein synthesis and degradation. As the sentence “Nrf1 is not a direct target of SREBP1, albeit its indirect effects on upstream signaling to Nrf1 cannot be ruled out”mentioned, we have no evidence that SREBP1 directly acts on Nrf1, but P97, DDI1, DDI2 and Hrd1, as upstream signals of Nrf1, are affected differently with siSREBP1’s effects, so that it cannot be ignored. Overall, thank you again for your constructive suggestions, enabling careful revision to improve the quality of this paper.

Reviewer #2's Comments

Thank you very much for your careful review of this manuscript. The issues you pointed out are very meaningful and professional for us. In response to your comments and questions, we have made the following answers.

POINT 1: Nrf1's processing protein DDI1/2 increase when SREBP1 is knocked down, but Nrf1 does not change, has there been a transform in the nucleocytoplasmic distribution of Nrf1? 

Response: Thank you very much for your comments and thank you for your valuable suggestions. When SREBP1 is knocked down, though Nrf1's processing protein DDI1/2 increased, abundances of all Nrf1 isoforms were marginally enhanced and there has no transform in the nucleocytoplasmic distribution of Nrf1. 

POINT 2: DDI1 knockout affects the expression of Nrf1 and Nrf2, but there is no change in tumor growth. Is this related to the simultaneous changes in Nrf1 and Nrf2?

Response: Thank you very much for your suggestion, which is very fundamental for us. Yes, we think it's the combination of Nrf1 and Nrf2, which have a close interaction in the occurrence and development of liver cancer and tumor growth, which have opposite effects in regulating tumor growth particularly. When DDI1 knockout, both the expression of Nrf1 and Nrf2 were decreased, so that's probably why the tumor hasn't changed significantly.

POINT 3: Is the effect of DDI1 deficiency on the reactive signaling pathway of rapamycin to proteasomes also regulated by the expression of other proteasome subunits?

Response: Thank you very much for your questions. We think that DDI1 deficiency has effected on the reactive signaling pathway of rapamycin to proteasomes, but is not regulated by the expression of other proteasome subunits.

POINT 4: Please check the text for minor errors, e.g., in Figure 5A has an extra letter 'A' above it.

Response: We apologize to the reviewer for the errors in the old version and thank you very much for rigorous and precise examination, and for helping me find and revise the minor errors in the text. In our updated text, we have made timely corrections, which are changed font format in page 3,4,5 and an extra letter 'A' above Figure 5A removed and we will re-upload Figure 5.

POINT 5: (Fig. 4F and Fig. 5) The resolution of some photographs is too low. Please improve them.

Response: Thank you for your constructive comments promoting us to improve the text content. Now, we have further improved Figs. 4F and 5 with this revised manuscript being uploaded.

Overall, we are thankful to you again for your patient constructive comments and suggestions, as well as thanks for your positive comments, which is quite important for us. Thus, we have improved the revision of the text with your help, and it is clearly visible in the uploaded version. Thank you for your constructive comments urging us to further revise and improve the text content.

---

## [Decision Letter · Decision Letter 1]

3 Nov 2023

Nrf1 is not a direct target gene of SREBP1, albeit both are integrated into the rapamycin-responsive regulatory network in human hepatoma cells

PONE-D-23-03948R1

Dear Dr. Zhang,

We’re pleased to inform you that your manuscript has been judged scientifically suitable for publication and will be formally accepted for publication once it meets all outstanding technical requirements.

Kind regards,

Salvatore V Pizzo

Academic Editor

PLOS ONE

Additional Editor Comments (optional):

Reviewers' comments:

Reviewer's Responses to Questions

**Comments to the Author**

1. If the authors have adequately addressed your comments raised in a previous round of review and you feel that this manuscript is now acceptable for publication, you may indicate that here to bypass the “Comments to the Author” section, enter your conflict of interest statement in the “Confidential to Editor” section, and submit your "Accept" recommendation.

Reviewer #2: All comments have been addressed

2. Is the manuscript technically sound, and do the data support the conclusions?

Reviewer #2: Partly

3. Has the statistical analysis been performed appropriately and rigorously? 

Reviewer #2: Yes

4. Have the authors made all data underlying the findings in their manuscript fully available?

Reviewer #2: Yes

5. Is the manuscript presented in an intelligible fashion and written in standard English?

Reviewer #2: Yes

6. Review Comments to the Author

Reviewer #2: (No Response)

7. PLOS authors have the option to publish the peer review history of their article (what does this mean?). If published, this will include your full peer review and any attached files.

Reviewer #2: No

---

## [Editor Report · Acceptance letter]

13 Nov 2023

PONE-D-23-03948R1 

Nrf1 is not a direct target gene of SREBP1, albeit both are integrated into the rapamycin-responsive regulatory network in human hepatoma cells 

Dear Dr. Zhang:

I'm pleased to inform you that your manuscript has been deemed suitable for publication in PLOS ONE. Congratulations! Your manuscript is now with our production department. 

Kind regards, 

on behalf of

Dr. Salvatore V Pizzo 

Academic Editor

PLOS ONE